# Remediation of Lead Contamination by *Aspergillus niger* and Phosphate Rocks under Different Nitrogen Sources

Yi Feng [1,2,3,4], Liangliang Zhang [1,2,3,4], Xiang Li [1,2,3,4], Liyan Wang [1,2,3,4], Kianpoor Kalkhajeh Yusef [1,2,3,4], Hongjian Gao [1,2,3,4] and Da Tian [1,2,3,4],*

1   Anhui Province Key Lab of Farmland Ecological Conservation and Pollution Prevention, School of Resources and Environment, Anhui Agricultural University, Hefei 230036, China; fengyi@stu.ahau.edu.cn (Y.F.); zhangliangliang@ahau.edu.cn (L.Z.); 869684757@stu.ahau.edu.cn (X.L.); 19710023@stu.ahau.edu.cn (L.W.); yusef.kianpoor@hotmail.com (K.K.Y.); hjgao@ahau.edu.cn (H.G.)
2   Research Centre of Phosphorus Efficient Utilization and Water Environment Protection along the Yangtze River Economic Belt, Anhui Agricultural University, Hefei 230036, China
3   Anhui Engineering and Technology Research Center of Intelligent Manufacture and Efficient Utilization of Green Phosphorus Fertilizer, Anhui Agricultural University, Hefei 230036, China
4   Key Laboratory of JiangHuai Arable Land Resources Protection and Eco-Restoration, Ministry of Natural Resources, Hefei 230036, China
*   Correspondence: tianda@ahau.edu.cn

**Abstract:** Co-application of *Aspergillus niger* (*A. niger*) and phosphate rocks (PR) has been practiced by environmentalists for lead (Pb) remediation. The secretion of organic acid by *A. niger* usually dominates the dissolution of PR and Pb immobilization. In this study, two types of PR (fluorapatite (FAp) and phosphogypsum (PG)) were investigated in Pb remediation by *A. niger* under three different forms of nitrogen (ammonium, nitrate, and urea). Our results reveal that the formation of pyromorphite and lead oxalate contributed to Pb removal by the combination of *A. niger* with FAp and PG. PG showed a significant capability for Pb remediation compared with FAP, over 94% of Pb vs. 50%. Compared with nitrate and urea, ammonium significantly decreased Pb cation concentrations from 1500 mg/L to 0.4 mg/L. Due to ammonium containing sulfate, the lead sulfate formed also contributed to Pb removal. However, nitrate stimulated *A. niger* to secrete more oxalic acid (~1400 mg/L) than ammonium and urea (~200 mg/L), which can form insoluble lead oxalate. These insoluble minerals can reduce the availability of removed Pb. Despite the efficacy of both ammonium and nitrate for Pb remediation, our findings suggest that nitrate is the primary candidate in this regard due to high oxalic acid secretion.

**Keywords:** *Aspergillus niger*; fluorapatite; nitrogen; phosphogypsum; Pb remediation

## 1. Introduction

The problem of environmental pollution caused by lead (Pb) is worldwide [1,2]. Human activities are the main causes of Pb discharge into the environment, such as urban and industrial wastes, mining, and chemical emissions [3–5]. Pb contamination in the environment can directly threaten human health through a variety of exposure pathways [6,7]. Even at a low level (10 ug/dL), exposure to Pb can cause health problems, including loss of hearing, chronic brain injury, high blood pressure, etc., especially in children [6,8,9]. Due to its non-biodegradability, Pb can gradually accumulate in the environment and cause human toxicity through food consumption [10,11]. Hence, it is of major importance to remediate Pb contaminated sites.

The utilization of phosphate is an effective way to immobilize Pb in water, soil, and waste [2]. Phosphate has a high affinity for Pb, with an adsorption capacity of up to 138 mmol/kg [12]. More importantly, phosphorus (P) in phosphate can react with Pb to form highly insoluble pyromorphite (Pyro, $Pb_5(PO_4)_3Cl$, OH, F) [10,13,14]. $Cl^-$ ions

easily exist in polluted environments, and $Pb_5(PO_4)_3Cl$ is the most stable product in the apatite–lead remediation system [13]. Fluorapatite (FAp) is the most common phosphate rock in nature that is highly effective for Pb remediation via the formation of Pyro [10,14,15]. Pyro formation in solution contributes to over 99% of Pb immobilization [10]. Meanwhile, phosphogypsum (PG) is the main by-product of the wet production of phosphoric acid containing ~15% water-soluble P, which can directly react with Pb to form Pyro [16,17]. Nevertheless, most P exists in PG as insoluble phosphate [18,19]. Similar to FAp, PG phosphate has a low solubility ($Ksp < 10^{-70}$) and limits Pb remediation [20]. Therefore, it is essential to stimulate P release from FAp and PG for efficient Pb remediation.

Phosphate-solubilizing microorganisms (PSMs) can improve the dissolution of insoluble phosphates (IPs) [17,20,21]. Compared with other PSMs (e.g., bacteria), phosphate-solubilizing fungi (PSF) have shown a stronger ability (~10 times) for P release from IPs in solid and liquid media [22,23]. Additionally, PSF has a high tolerance for Pb toxicity [10,21]. For example, *Aspergillus niger* (*A. niger*) and *Penicillium oxalicum* (*P. oxalicum*) can respectively resist 1000 and 1500 mg/L Pb levels [16]. The combination of these two fungi and FAp can effectively remove > 98% of Pb cations via the formation of Pyro and lead oxalate [10,20].

Oxalic acid is the primary organic acid secreted by PSF [24,25]. Compared with other organic acids, oxalic acid has the highest acidity constant (pKa1 = 1.25 and pKa2 = 4.27), which can significantly decrease the medium pH and dissolve IPs [22]. On top of its strong chemical acidity (e.g., compared to sulfuric acid), oxalic acid is more efficient for the solubilization of rock phosphate [26]. Oxalic acid secreted by *A. niger* not only promotes the dissolution of phosphate rock, but also contributes to Pb immobilization via the formation of Pb oxalate, suggesting that the secretion of oxalic acid by *A. niger* dominates the process of Pb remediation in water by phosphate [10,21,27,28].

The secretion of oxalic acid by PSF is affected by different environmental factors, such as carbon (C) source, nitrogen (N) source, and environmental pH [29]. Nitrogen is a key factor affecting the metabolism of *A. niger*, and the solubilization of phosphate rocks [30]. Therefore, the existence of nitrogen will affect Pb availability and its removal in the presence of *A. niger* and phosphate (FAp and PG). Ammonium nitrogen can effectively promote the release of P from IPs, ~7 and ~60 times higher than nitrogen and urea, respectively [30]. However, the secretion of oxalic acid by *A. niger* is stimulated by the presence of nitrate [31]. Compared with ammonium and urea, nitrate should be more efficient in Pb remediation by *A. niger* and phosphate via the formation of lead oxalate and Pyro. Hence, selecting a suitable N source is a prerequisite for the remediation of Pb by *A. niger* and phosphate. In addition, the bioavailability of removed Pb cations in the PSF–phosphate system is also unclear.

It was hypothesized that nitrogen sources (ammonium, nitrate, and urea) could affect Pb availability and its removal in the presence of phosphate (FAp and PG). This study aimed to explore the influence of different N sources (ammonium, nitrate, and urea) on Pb remediation by *A. niger* and phosphate (FAp and PG). The bioavailabilities of removed Pb in the precipitates were also detected.

## 2. Materials and Methods

### 2.1. FAp, PG, and Fungal Strains Preparation

FAp rock was collected from Kaiyang Phosphate Rock Reserve in Guizhou Province (N 27°6′50″, E 106°51′5″), China. All the FAp samples were ground to powder, dried, and filtered through a 150-μm mesh sieve. PG was collected from the phosphorus chemical industry in Tongling City, Anhui Province, China. The collected PG was air-dried and passed through a 150-μm mesh sieve [18].

*Aspergillus niger* (AH-F-1-2, CGMCC No. 23272) were isolated from soybean rhizospheres at the North Anhui comprehensive experimental station in Suzhou City, Anhui Province, China. *Aspergillus niger* was cultured in Potato Dextrose Agar (PDA) medium at 28 °C for five days, and then drenched with sterile water. Afterwards, mycelium fragments were removed with three layers of sterile cheesecloth to obtain the spore suspensions. Fi-

nally, the filtrate was diluted with sterile water and the spore concentration was measured by a hemocytometer and adjusted to $10^7$ CFU/mL [18,22].

### 2.2. Pb Remediation by A. niger and Phosphate under Different Nitrogen

The Pb contamination in water was prepared by lead acetate basic ($C_4H_8O_6Pb_2$) powder (Shanghai Macklin Biochemical Co., Ltd., Shanghai, China). The initial Pb concentration in water was adjusted to 1500 mg/L. FAp and PG were added to Pikovskaya (PVK) liquid medium as inorganic P resources before incubation, which contained 10 g glucose, 0.2 g NaCl, 0.25 g $MgSO_4·7H_2O$, 0.03 g $MnSO_4·4H_2O$, and 0.2 g KCl in 1 L deionized water (China National Pharmaceutical Group Chemical Reagent Co., Ltd., Shanghai, China). Three forms of N were added to the PVK medium, i.e., ammonium ($(NH_4)_2SO_4$), nitrate ($Ca(NO_3)_2·4H_2O$), and urea ($CO(NH_2)_2$). The inoculation amount of N was 0.5 g in the medium.

The experiment was performed with nine treatments, i.e., ANG (*A. niger* only), ANG + FAp (*A. niger* + FAp), ANG + $NH_4^+$ + FAp (*A. niger* + $(NH_4)_2SO_4$ + FAp), ANG+ $NO_3^-$ + FAp (*A. niger* + $Ca(NO_3)_2.4H_2O$ + FAp), ANG + urea + FAp (*A. niger* + urea + FAp), ANG + PG (*A. niger* + PG), ANG + $NH_4^+$ + PG (*A. niger* + $(NH_4)_2SO_4$ + PG), ANG + $NO_3^-$ + PG (*A. niger* + $Ca(NO_3)_2.4H_2O$ + PG), and ANG + urea + PG (*A. niger* + urea + PG). Before the incubation, 0.205 g $C_4H_8O_6Pb_2$ (Pb,1500 mg/L) was added to all conical flasks as the Pb contaminants. Then, 1 g FAp and 1 g PG were added to 250 mL conical flasks with a 100 mL PVK culture medium, respectively. Subsequently, the PVK medium was sterilized in the autoclave at 121 °C for 20 min. Finally, 1 mL of spore suspension was added to each conical flask. The parafilm (BS-QM-003, Biosharp) containing a 0.22 μm gas-permeable membrane in the middle was used to seal the conical flasks. All operations were carried out under sterile conditions.

After incubation for two, four, and six days at 180 rpm, 28 °C, the PVK medium was filtered with P-free filter paper to collect filtrate and precipitate in each treatment. The collected filtrate was filtered through a 0.22 μm polyethersulfone (PES) membrane for testing pH, organic acid, P, and Pb concentrations. The collected precipitates were dried for 24 h at 65 °C for the analysis of dry biomass, XRD, and SEM–EDS. All treatments were conducted with three replicates.

### 2.3. TCLP-Pb Extraction from Precipitates

The TCLP method was used to extract the leached Pb content in precipitates. 2 g principles and 40 mL extractant were mixed and shaken at 180 rpm for 18 h (25 °C). Then, the solution was collected and filtered with a 0.22 μm polyethersulfone (PES) membrane for testing TCLP-Pb by ICP-OES [32]. Two extractants were prepared for the extraction of TCLP-Pb according to the pH. Extraction agent 1 (pH < 5): dissolve 5.7 mL of glacial acetic acid in 500 mL of deionized water, then add 64.3 mL of 1 mol/L NaOH, then dilute to 1 L, use 1 mol/L $HNO_3$ or 1 mol/L NaOH to adjust the pH of the solution within $4.93 \pm 0.05$ [28,33,34]. Extraction agent 2 (pH > 5): dissolve 5.7 mL of glacial acetic acid into 500 mL of deionized water, dilute to 1 L, and adjust the pH of the solution with 1 mol/L $HNO_3$ or 1 mol/L NaOH within $2.88 \pm 0.05$ [28,33,34].

### 2.4. Instrumentation

pH values were measured using an SG98 In Lab pH meter (Mettler Toledo Int. Inc., Columbus, OH, USA) with an Expert Pro-ISM-IP67 probe. Pb concentrations and TCLP-Pb concentrations were analyzed by inductively coupled plasma optical emission spectrometry (ICP-OES, Agilent 710). The concentration of calibration standards of P and Pb were 5, 10, 20, 50, and 100 mg/L.

The contents of the organic acids were determined by high-performance liquid chromatography (HPLC) (Agilent 1200, Agilent Technologies, Santa Clara, CA, USA) with a column temperature of 30 °C. Then, the standard solution of oxalic acid was prepared and diluted into 1000, 500, 100, 50, 20, and 10 mg/L, respectively. The R squared value of the

internal standard curve was 0.999. The mobile phase was prepared with 0.1% phosphoric acid ($H_3PO_4$) and methanol ($CH_3OH$) in a ratio of 99:1.

Mineralogical characterization of the crystallization products was examined by D/Max-2500 X-ray diffraction (XRD, Rigaku Corporation) (Cu-K$\alpha$; 36 kV; 20 mA; scanned from 5° to 60° at a speed of 4°/s). The XRD patterns were analyzed by MDI Jade 6.5 for phase identification. Before the XRD analysis, the dried precipitates were ball-milled in a planetary ball mill (Mitr YXQM, Changsha Mitrcn Instrument Equipment Co., Ltd. (MITR), Changsha, China) for 10 minutes and passed through a 100 μm-mesh sieve [18]. The observation of morphology and composition of minerals were analyzed by SEM (S4800 Hitachi) with an acceleration voltage of 7 kV. To enhance image quality and minimize charging, the samples were coated with gold for one minute in Hitachi E-1010 Sputter for SEM analysis. The semi-quantity analysis (collecting time: 90 seconds) was performed by Oxford Aztec X-Max 150 energy dispersive spectrometer (EDS).

### 2.5. Statistical Analysis

Flask experiments were performed in triplicate. The means and standard deviations in each treatment were calculated and presented. The significant differences among the treatments were identified by Tukey's honestly significant difference test ($p < 0.05$) via one-way ANOVA. XRD and SEM analyses were carried out on a range of samples; the results are presented in this study.

### 3. Results

### 3.1. Fungal Biomass, pH, and Oxalic and Citric Acid Secreted by A. niger

In ANG treatment, *A. niger* showed low biomass (from 0.04 to 0.10 g) during incubation (Figure 1). The dry biomass in ANG + FAp, ANG + $NH_4^+$ + FAp, ANG + $NO_3^-$ + Fap, and ANG + urea + FAp treatments significantly increased to 0.99, 1.01, 1.06, and 1.28 g, respectively, on day two (Figure 1A). The biomass in ANG + FAp had no evident changes on days four and six, i.e., 1.06 and 1.04 g, respectively (Figure 1A). Similar tendency appeared for ANG + $NO_3^-$ + FAp biomass on days four (1.24 g) and six (1.01 g) (Figure 1A). Compared with ANG + FAp and ANG + $NO_3^-$ + FAp treatments, the dry biomasses in ANG + $NH_4^+$ + FAp and ANG + urea + FAp treatment were significantly higher (1.64 and 1.31 g) on day six (Figure 1A). Likewise, the dry biomasses of ANG + PG, ANG + $NH_4^+$ + PG, ANG + $NO_3^-$ + PG, and ANG + urea + PG treatments increased markedly to 0.87, 1.00, 0.69, and 0.83 g, respectively, on day two, whereas those of ANG + PG and ANG + $NO_3^-$ + PG had no evident changes on day four (0.81 and 0.67 g) and six (0.76 and 0.83 g) (Figure 1B). However, substantial increases occurred in the dry biomasses of ANG + $NH_4^+$ + PG and ANG + urea + PG treatments on days four (1.56 and 1.38 g) and six (1.52 and 1.38 g) (Figure 1B).

The initial pH of the PVK medium was 6.5. After two days of incubation, the pH in ANG, ANG + FAp, ANG + $NH_4^+$ + FAp, ANG + $NO_3^-$ + FAp, and ANG + urea + FAp treatments were 4.5, 4.9, 5.1, 4.9, and 4.9, respectively (Figure 2A). In ANG, ANG + FAp, and ANG + $NO_3^-$ + FAp treatments, the pH values were similar on days four and six (Figure 2A). However, the pH in ANG + $NH_4^+$ + FAp treatment significantly increased up to 5.8 and 5.4 after four and six days of incubation, respectively (Figure 2A). In ANG + urea + FAp treatment, the pH was lower within days four to six, i.e., 4.0 and 4.5, respectively (Figure 2A). The pH in ANG + PG and ANG + $NO_3^-$ + PG had no evident changes on days two and six, i.e., 5.67 and 5.75, respectively, after six days of incubation (Figure 2B). In ANG + $NH_4^+$ + PG treatments, the pH values were 3.6, 3.5, and 4.7 after two, four, and six days of incubation, respectively (Figure 2B). In ANG + urea + PG treatment, the pH reduced to 4.0 after two days of incubation and then increased to 6.97 and 6.71 on days four and six, respectively (Figure 2B).

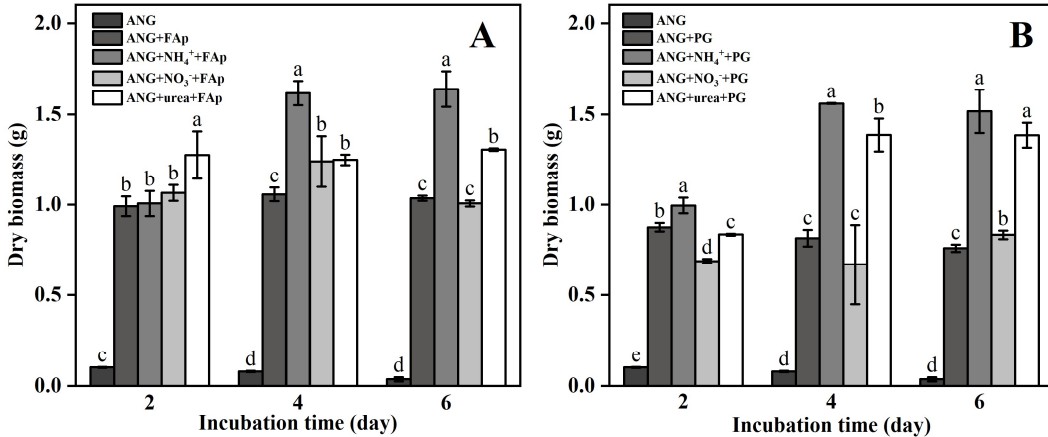

**Figure 1.** The dry biomass in the FAp (**A**) and PG (**B**) treatments during the incubation (2, 4, and 6 days). The error bars represent the standard deviations of three replicates. The different lower-case letters indicate a significant difference between the treatments ($p < 0.05$). The significant differences among the treatments were identified by Tukey test ($p < 0.05$) via one-way ANOVA.

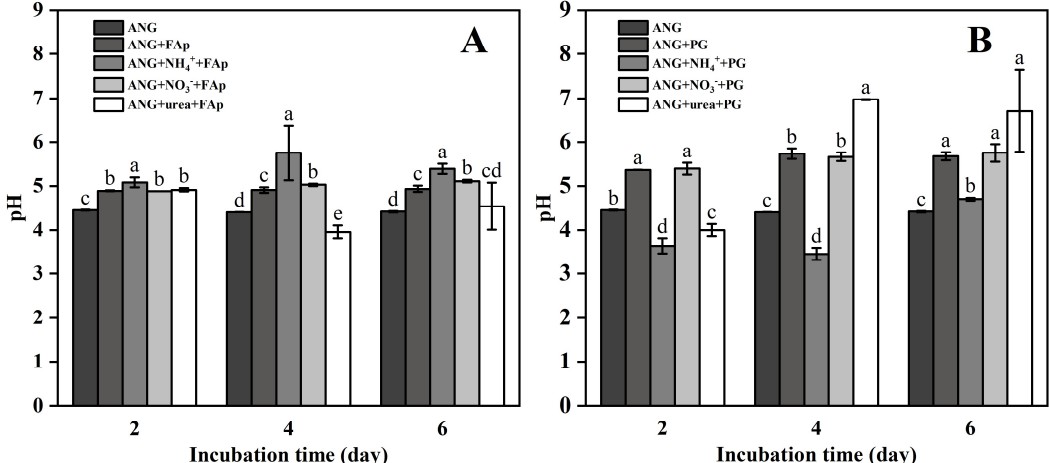

**Figure 2.** The pH value in adding the FAp (**A**) and PG (**B**) treatments during the incubation (2, 4, and 6 days). The error bars represent the standard deviations of three replicates. The different lower-case letters indicate a significant difference between the treatments ($p < 0.05$). The significant differences among the treatments were identified by Tukey test ($p < 0.05$) via one-way ANOVA.

The primary organic acids secreted by *A. niger* were oxalic acid and citric acid. In ANG treatment, the concentrations of oxalic acid were 267, 264, and 283 mg/L on days two, four, and six, respectively (Figure 3). The concentrations of oxalic acid in ANG + $NO_3^-$ + FAp treatment were 1379, 1375, and 1385 mg/L on days two, four, and six, respectively (Figure 3A). However, the concentrations of oxalic acid in ANG + FAp, ANG + $NH_4^+$ + FAp, and ANG + urea + FAp treatments were significantly lower than in ANG + $NO_3^-$ + FAp treatment during the incubation, i.e., ranged from 163 to 201 mg/L (Figure 3A). In addition, the highest concentrations of oxalic acid occurred in ANG + $NO_3^-$ + PG treatment, i.e., 1377, 1379, and 1387 mg/L after two, four, and six days of incubation (Figure 3B). In ANG + PG, ANG + $NH_4^+$ + PG, and ANG + urea + PG treatments, the concentrations of oxalic acid ranged from 218 to 193, 190 to 159, and 216 to 64 mg/L during the incubation, respectively (Figure 3B).

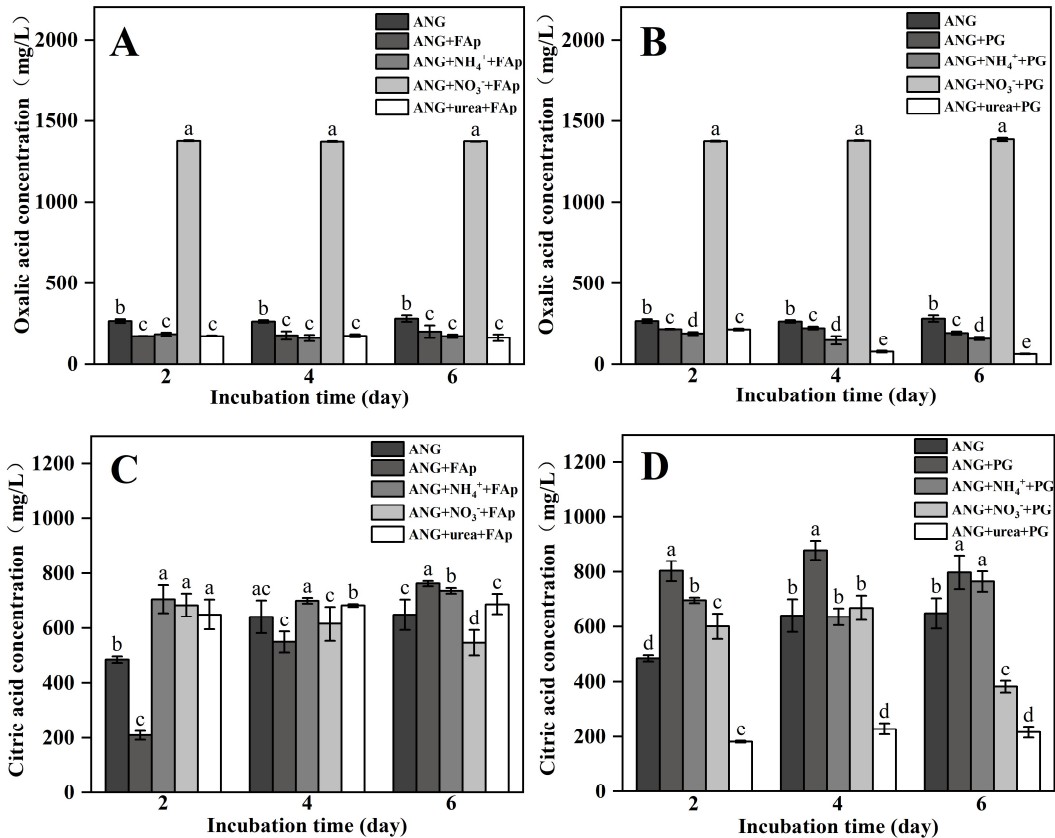

**Figure 3.** The secretion of oxalic acid and citric acid by ANG in the FAp (**A**,**C**) and PG (**B**,**D**) treatments during the incubation time (2, 4, and 6 days). The error bars represent the standard deviations of three replicates. The different lower-case letters indicate a significant difference between the treatments ($p < 0.05$). The significant differences among the treatments were identified by Tukey test ($p < 0.05$) via one-way ANOVA.

In the ANG treatment, citric acid increased from 486 to 641 and 649 mg/L on days two and six, respectively (Figure 3C,D). The concentrations of citric acid in ANG + FAp had an identical trend, i.e., 209, 550, and 763 mg/L on days two to six (Figure 3C). Similar concentrations occurred for citric acid in ANG + $NH_4^+$ + FAp and ANG + urea + FAp treatments after six days of incubation, i.e., 736 and 687 mg/L, respectively (Figure 3C). In the ANG + $NO_3^-$ + FAp treatment, the citric acid concentration decreased from 684 to 574 mg/L during the incubation (Figure 3C). In ANG + PG treatment, the concentrations of citric acid were 803, 879, and 798 mg/L on days two, four, and six, respectively, while those of ANG + $NH4^+$ + PG increased from 697 to 637 and 765 mg/L after two to six days of incubation (Figure 3D). In ANG + $NO_3^-$ + PG treatment, the concentration of citric acid on days two and four were 602 and 669 mg/L, respectively, and then decreased to 381 mg/L on day six (Figure 3D). Citric acid had lower concentrations in ANG + urea + PG treatment after two, four, and six days of incubation, i.e., 181, 228, and 216 mg/L, respectively (Figure 3D).

### 3.2. Pb and TCLP-Pb Concentrations

In ANG treatment, Pb concentrations were 1295, 1316, and 1355 mg/L on days two, four, and six (Figure 4). In both ANG + FAp and ANG + $NO_3^-$ + FAp treatments, Pb concentrations decreased to similar values of ~700 mg/L during the incubation time (Figure 4A). Significant reductions occurred in Pb concentrations of ANG + $NH_4^+$ + FAp and ANG + urea + FAp treatments from 376.9 and 658.6, respectively, to 12.5 and 43.8 mg/L, respectively, after six days of incubation (Figure 4A). In ANG + PG, ANG + $NH_4^+$ + PG, ANG + $NO_3^-$ + PG, and ANG + urea + PG treatments, Pb concentrations were lower after two days of incubation, ranging from 213.5 to 33.5 mg/L (Figure 4B). The minimum

value of Pb concentration in ANG + NH$_4^+$ + PG treatment was 0.4 mg/L after six days of incubation (Figure 4B).

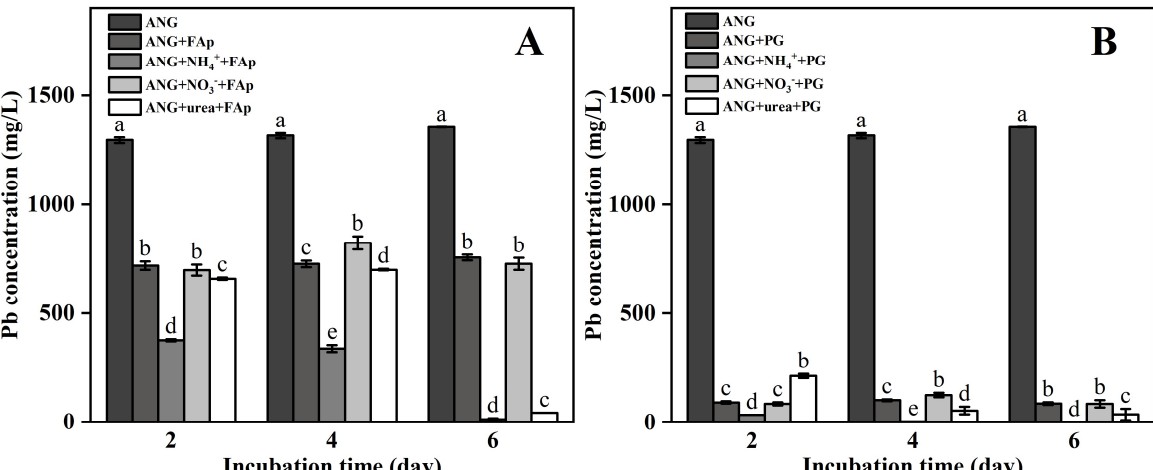

**Figure 4.** Concentrations of Pb between FAp (**A**) and PG (**B**) treatments during the incubation (2, 4, and 6 days). The error bars represent the standard deviations of three replicates. The different lower-case letters indicate a significant difference between the treatments ($p < 0.05$). The significant differences among the treatments were identified by Tukey test ($p < 0.05$) via one-way ANOVA.

The concentrations of TCLP-Pb in ANG treatment were 3705, 3074, and 3145 mg/L on days two, four, and six, respectively (Figure 5). In ANG + FAp, ANG + NH$_4^+$ + FAp, ANG + NO$_3^-$ + FAp, and ANG + urea + FAp treatments, the concentrations of TCLP-Pb decreased to 294, 469, 297, and 284 mg/L after two days of incubation (Figure 5A). In ANG + NH$_4^+$ + FAp treatment, the TCLP-Pb concentration continuously decreased to 263 and 23 mg/L on days four and six, respectively (Figure 5A). In contrast, TCLP-Pb concentration increased to 374 mg/L on day four, and then decreased to 77 mg/L on day six, in ANG + urea + FAp treatment (Figure 5A). In ANG + PG, ANG + NH$_4^+$ + PG, ANG + NO$_3^-$ + PG, and ANG + urea + PG treatments, the concentrations of TCLP-Pb decreased to 24.8, 12.4, 52.7, and 36.4 mg/L, respectively, after two days of incubation (Figure 5B). In these treatments, lower concentrations of TCLP-Pb from 70.2 to 11.4 mg/L appeared after four and six days of incubation (Figure 5B).

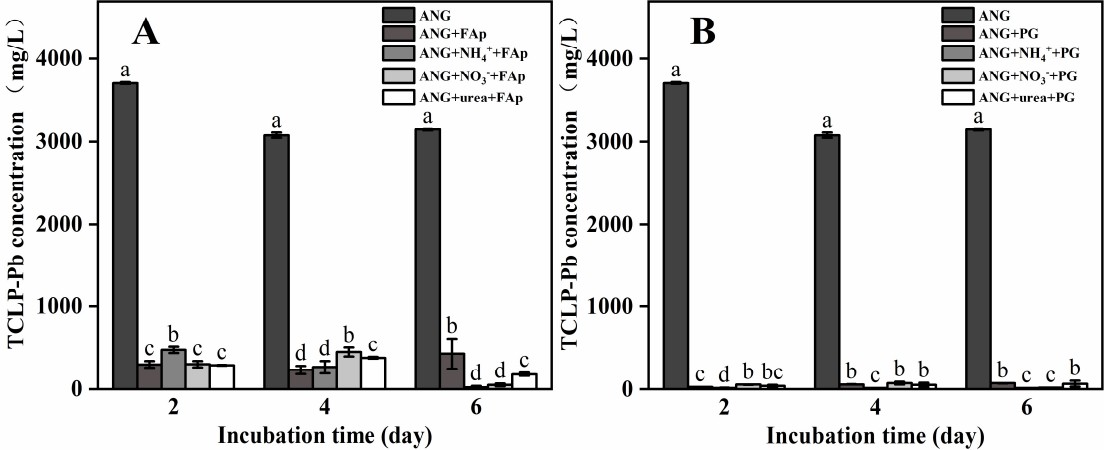

**Figure 5.** Concentrations of TCLP-Pb in the FAp (**A**) and PG (**B**) treatments during the incubation (2, 4, and 6 days). The different lower-case letters indicate a significant difference between the treatments ($p < 0.05$). The significant differences among the treatments were identified by Tukey test ($p < 0.05$) via one-way ANOVA.

## 3.3. XRD Analysis

Figure 6 shows the XRD patterns for each treatment after six days of incubation. The peaks in lead oxalate ($Pb(C_2O_4)$) were detected for all treatments at 20.1, 23.8, and 25.9° (Figure 6). In the FAp treatment, the weakly peaks in pyromorphite at 33.5 and 52.5° were also observed in all treatments (Figure 6A). However, the newly formed minerals of lead sulfate ($PbSO_4$, at 43.95°) only occurred in the ANG + $NH_4^+$ + FAp treatment (Figure 6A). In addition, the lead carbonate ($PbCO_3$, at 48.9°) also formed in the ANG + urea + FAp treatment (Figure 6A). In the PG treatment, the weakly peak in pyromorphite at 30.5° was also observed in ANG + $NH_4^+$ + PG and ANG + urea + PG (Figure 6B). In addition, the lead carbonate ($PbCO_3$, at 29.2°) appeared in ANG + urea + PG treatment (Figure 6B). The peak in lead sulfate ($PbSO_4$, at 43.9°) was observed in all treatments adding PG (Figure 6B).

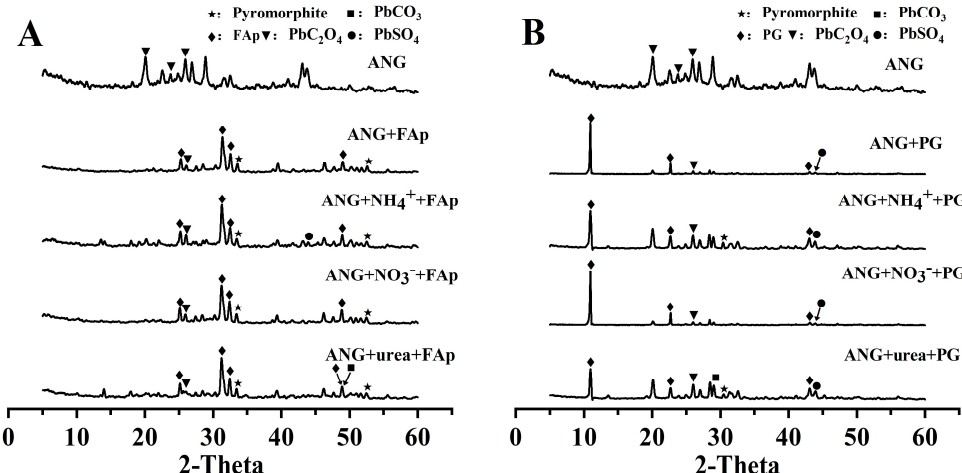

**Figure 6.** XRD patterns of precipitation in the FAp (**A**) and PG (**B**) treatments after six days of incubation.

## 3.4. SEM–EDS Analysis

Figure 7 shows the morphology of the minerals formed in the treatments of *A. niger* combined with FAp after six days of incubation. The hyphae of *A. niger* were observed in ANG + FAp, ANG + $NH_4^+$ + FAp, ANG + $NO_3^-$ + FAp, and ANG + urea + FAp treatments (Figure 7). EDS also indicated the existence of Pyro in ANG + FAp (Figure 7A) and lead oxalate mineral in ANG + $NH_4^+$ + FAp (Figure 7B). The calcium oxalate mineral was also identified in ANG + $NH_4^+$ + FAp treatment (Figure 7B). The $CaSO_4$ mineral only appeared in ANG + urea + FAp treatment (Figure 7D).

Figure 8 shows the morphology of the minerals formed in treatments of *A. niger* combined with PG after six days of incubation. The hyphae of *A. niger* and lead oxalate were observed in ANG + PG, ANG + $NH_4^+$ + PG, ANG + $NO_3^-$ + PG, and ANG + urea + PG treatments (Figure 8). This is consistent with the XRD results. However, lead oxalate was blocky in ANG + PG (Figure 8A) and ANG + $NO_3^-$ + PG (Figure 8C), flaky in ANG + $NH_4^+$ + PG (Figure 8B), and acicular in ANG + urea + PG (Figure 8D). EDS also indicated the existence of $Ca_3(PO_4)_3$ mineral in the ANG + PG treatment (Figure 8A).

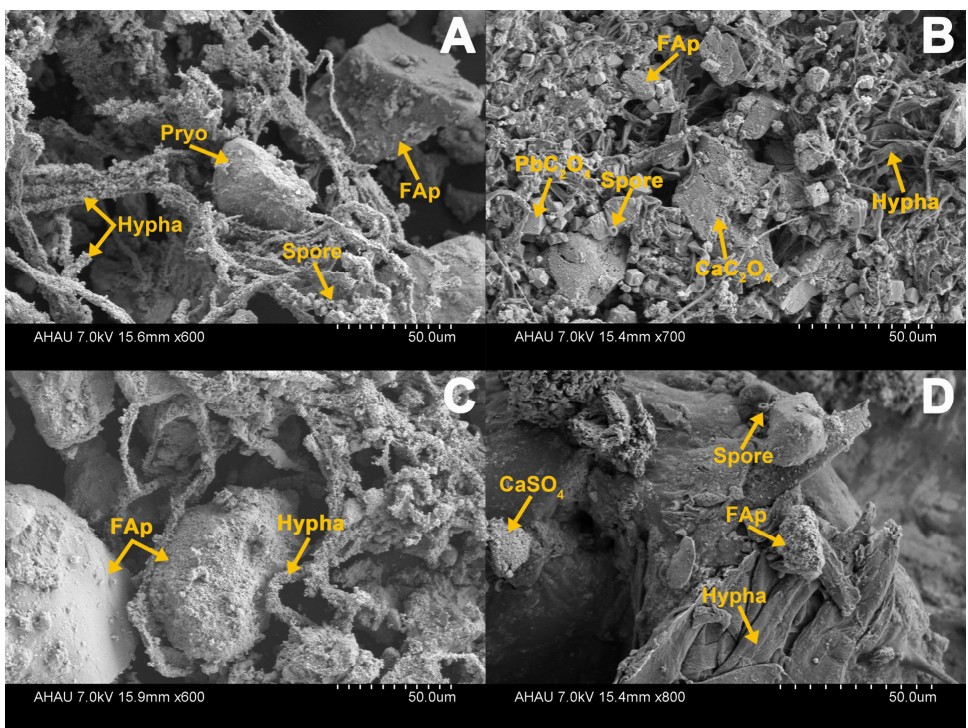

**Figure 7.** SEM image and EDS point data for ANG + FAp (**A**), ANG + $NH_4^+$ + FAp (**B**), ANG + $NO_3^-$ + FAp (**C**), and ANG + urea + FAp (**D**) treatments after six days of incubation.

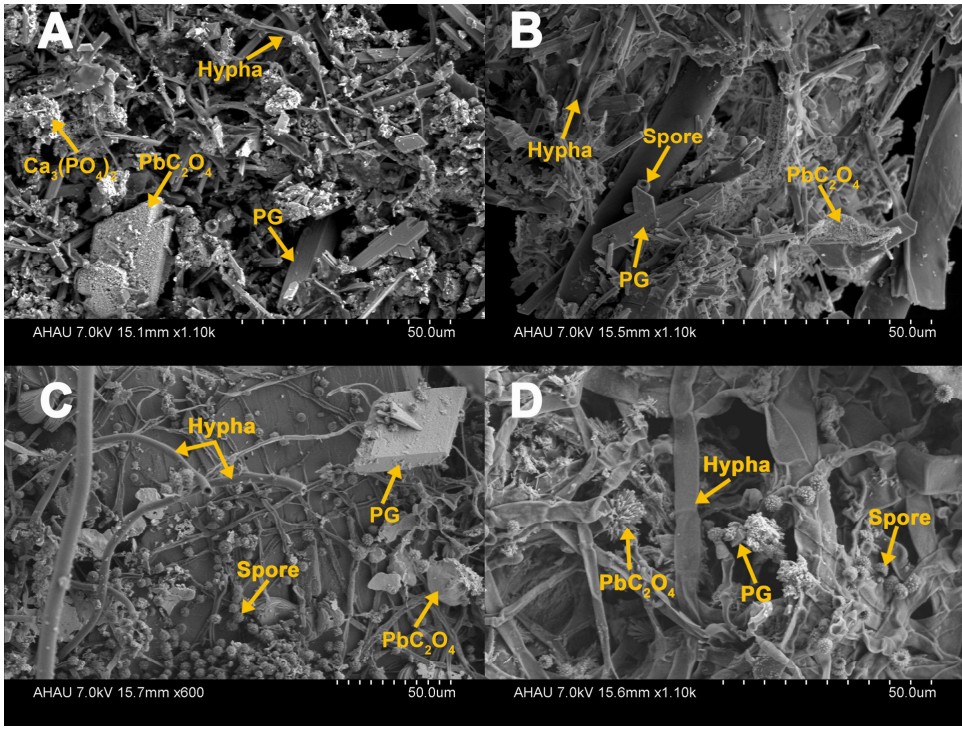

**Figure 8.** SEM image and EDS point data for ANG + PG (**A**), ANG + $NH_4^+$ + PG (**B**), ANG + $NO_3^-$ + PG (**C**), and ANG + urea + PG (**D**) treatments after six days of incubation.

## 4. Discussion

The combination of *A. niger* and phosphate has been successfully applied to Pb remediation in water [10,21]. On one hand, *A. niger* has a high Pb tolerance (up to 1500 mg/L), and is thereby highly efficient for Pb removal via adsorption and accumulation pathway [21,34,35].

On the other hand, the secretion of organic acid (e.g., oxalic acid) can react with Pb to form insoluble lead oxalate [21]. Oxalic acid can also contribute towards P releasing from phosphate and reacting with Pb to form highly insoluble Pyro [10,13,36,37]. Our results confirm that *A. niger* can survive under high Pb toxicity (1500 mg/L) and maintain the ability of organic acid secretion (Figures 1, 3, S2 and S3). Both FAp and PG can promote the growth of *A. niger* and function in Pb removal from the medium (Figures 1 and 4).

Although phosphate fertilizer is more effective at removing high concentrations of Pb, P is also easily immobilized by cations (e.g., $Ca^{2+}$, $Fe^{3+}$ and $Al^{3+}$) [38,39]. Therefore, the continued release of P is needed in Pb remediation. FAp is an efficient material in Pb remediation, and phosphate-solubilizing fungi can maintain the continuous release of P [10,14]. PG can also produce sustainable phosphate-based fertilizer by *A. niger* [17,40]. Unlike geological FAp, PG has an acidic pH (~2) and usually contains toxic elements (e.g., heavy metals, fluorine, etc.) [41,42]. These toxic elements not only cause environmental pollution, but also inhibit the metabolism of microorganisms [43]. However, both FAp and PG combined with *A. niger* can significantly promote Pb removal (Figure 4A,B). Compared with FAp, PG is more efficient for Pb removal by *A. niger*, i.e., 94% vs. 50% (Figure S1). Although PG has a lower P content (~2%), it releases P easier than FAp [17]. Released P can directly react with Pb to form highly insoluble Pyro at the beginning of incubation. Additionally, PG contains large amounts of $SO_4^{2-}$ that, in turn, can react with Pb to form insoluble lead sulfate. More importantly, the addition of FAp and PG can significantly decrease the TCLP-Pb concentration (Figure 5). Although *A. niger* can partly remove Pb cations in solution, the adsorbed Pb by *A. niger* could also be released in an acidic environment [21,33,44]. These results that phosphate is necessary for Pb remediation by *A. niger*, especially for the immobilization of available Pb.

Oxalic acid dominated the removal of Pb cations by *A. niger* combined with both FAp and PG. The formed minerals of lead oxalate were detected by XRD and observed by SEM–EDS (Figures 6–8). However, oxalic acid secretion was influenced by N sources [45]. The secretion of oxalic acid by *A. niger* is usually promoted by nitrate and restrained by ammonium [29]. Nitrate can promote the accumulation of oxalic acid via the inhibition of oxalic acid oxidase, hence it is beneficial to stimulate the secretion of oxalic acid by *A. niger* [46]. Our results also confirm that nitrate can significantly improve the secretion of oxalic acid by *A. niger* in Pb remediation (Figure 3). However, ammonium showed a higher Pb removal ratio compared with nitrate in FAp treatments, i.e., 99.2% vs. 52% (Figure S1A). Since ammonium (($NH_4$)$_2SO_4$) contains $SO_4^{2-}$, the lead sulfate formed in the ANG + $NH_4^+$ + FAp treatment contributed to the high Pb removal ratio (Figure S1A). In addition, the existence of $SO_4^{2-}$ in ammonium and PG also led to a similar Pb removal ratio between ANG + $NH_4^+$ + FAp and three PG treatments, i.e., 99.2% vs. 99.9% (Figure S1).

Ammonium can better promote the growth of *A. niger* compared to nitrate or urea [30], which our results confirm (Figure 1). The growth of *A. niger* is not only beneficial for phosphate dissolution and plant nutrition, but also it functions in Pb removal [47]. In addition, the decomposition of urea produces carbon dioxide and forms carbonates. The latter inhibit the growth of *A. niger* and the secretion of organic acids [48]. Hence ammonium and nitrate have higher efficiencies for Pb remediation by *A. niger* and phosphate than urea.

## 5. Conclusions

Our study investigated the co-application of *A. niger* and phosphate rocks for Pb remediation under different N conditions. Compared with *A. niger*, the addition of both FAp and PG significantly promoted the removal of Pb cations, reducing Pb availability. The formation of Pyro and lead oxalate contributed to Pb removal by the combined application of *A. niger* with FAp and PG. PG was more efficient than FAp for Pb remediation by *A. niger* via the formation of lead sulfate. Nitrate significantly promoted the production of oxalic acid by *A. niger* to a greater degree than ammonium and urea. Altogether, our results suggest the co-application of *A. niger* and PG, along with nitrate, as a cost-effective method for Pb remediation.

**Supplementary Materials:** The following supporting information can be downloaded at: https://www.mdpi.com/article/10.3390/agronomy12071639/s1. Figure S1. The Pb removal ratio in the FAp (A) and PG (B) treatments during the incubation time (2, 4, 6 days). Figure S2. The secretion of malic acid in the FAp (A) and PG (B) treatments during the incubation time (2, 4, 6 days). Figure S3. The secretion of acetic acid in the FAp (A) and PG (B) treatments during the incubation time (2, 4, 6 days).

**Author Contributions:** Conceptualization, D.T.; methodology, Y.F. and D.T.; validation, Y.F., L.Z. and X.L.; formal analysis, Y.F., L.Z., X.L. and L.W.; investigation, Y.F., L.Z., X.L. and L.W.; resources, D.T. and H.G.; data curation, D.T. and Y.F.; writing—original draft preparation, D.T. and K.K.Y.; writing—review and editing, D.T., Y.F. and K.K.Y.; supervision, D.T. and H.G.; funding acquisition, D.T. and H.G. All authors have read and agreed to the published version of the manuscript.

**Funding:** This research was funded by the program at the Department of Natural Resources of Anhui Province (NO. 2021-K-11 and NO. 2021-K-4), the National Natural Science Foundation of China (NO. 42007030 and NO. 41877099), the Natural Science Foundation of Anhui Province (NO. 2008085QD187), and the program at Anhui Agricultural University (NO. yj2019-20).

**Institutional Review Board Statement:** Not applicable.

**Informed Consent Statement:** Not applicable.

**Data Availability Statement:** Not applicable.

**Acknowledgments:** We appreciated Yang Xu from the Biotechnology Center of Anhui Agricultural University for the technical support of SEM in this study.

**Conflicts of Interest:** The authors declare no conflict of interest.

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
