# Peer review of "Remediation of Lead Contamination by Aspergillus niger and Phosphate Rocks under Different Nitrogen Sources"

_agronomy, doi:10.3390/agronomy12071639_

Round 1

Reviewer 1 Report

All of my comments have been addressed and the manuscript is ready to publish. 

Author Response

COMMENTS TO THE AUTHOR:

Review 1:All of my comments have been addressed and the manuscript is ready to publish.

Response:  Thank you very much for your comments. The language and minor spell has been checked and modified.

Reviewer 2 Report

The work hits one of the most interesting topics of recent years. In the reviewer's opinion, the article is already in a correct form for publication, except to correct some formatting errors:

- why are some words or parts of the text in blue?

- line 93: there is a double space between lines 93 and 94.

Author Response

COMMENTS TO THE AUTHOR:

Review 2:The work hits one of the most interesting topics of recent years. In the reviewer's opinion, the article is already in a correct form for publication, except to correct some formatting errors:

      Response: Thank you very much for your comments. The language and minor spell has been ckecked and moified.

  1. why are some words or parts of the text in blue?

     Response:Yes,the bule words or parts were the modified version (In PDF version). The manuscript also has been uploaded the clean version (in word).

     2. line 93: there is a double space between lines 93 and 94.

     Response:Has been corrected.

This manuscript is a resubmission of an earlier submission. The following is a list of the peer review reports and author responses from that submission.

Round 1

Reviewer 1 Report

This is a detailed study on the effect of N presence on the effectiveness of Pb immobilization by phosphates stimulated by microorganisms. Unfortunately, lack of understanding of thermodynamic principles of the phenomenon and the mechanisms of the reaction prevented the authors from proper design of the experiments and proper interpretation. The authors concentrated their work on organic processes ignoring completely all the inorganic side of the reaction, apparently competing for the effectiveness. There is lack of proper blanks and controls in the experiments and the background explanation and the conclusions are partial which does not move our knowledge ahead.

Therefore, in my opinion, this manuscript should be rejected.

The following specific comments have been made by the reviewer addressing selected, most important issues: omissions in the introduction, incorrect experimental setup, incomplete treatment of the result, numerous language issues, lack of proper controls and blanks.

Introduction and literature cited:

The authors used selected literature and the selection criteria are not entirely clear. There is a lack of citation of literature related to the discovery, patenting and implementation of the phosphate-induced Pb immobilization phenomenon (see for example papers by Ma, Traina, Manecki, etc.). Therefore, it is clear from the Introduction that the authors do not understand the mechanism of the phenomenon. Phosphate-induced immobilization of Pb involves the induction of pyromorphite precipitation by the addition of a P source in the presence of Cl ions. Cl ions are always present in the environment, particularly in the contaminated environment. The precipitated pyromorphite Pb5(PO4)3Cl is the most stable form of this system (see papers by Nriagu) and there is no chance for fluorpyromorphite to form in the environment, even when the source of P is fluorapatite. Formation of thermodynamically stable pyromorphite always outcompetes fluorpyromorphite. Lack of understanding of the basics of the method resulted in an incorrect experimental setup. This can no longer be corrected because you would have to do the experiment all over again. However, this should be acknowledged in the introduction and taken into account when describing the experiments and interpreting the results.

The experimental setup:

1) The concentration of Pb equal to 1.5 g/L in solution is unrealistic. Lead in contaminated soil and waste exists mostly in particulate form for a reason: because all Pb-containing species are nearly insoluble in water.  Besides, it is much cheaper and more effective to remove the high concentration of Pb simply by phosphate fertilizers, no PSF is needed. PSF is needed for prolonged removal of low levels of Pb using slowly-releasing sources of P, for example fluorapatite. Therefore, this is redundant to run experiments with that high Pb level.

2) Cl (KCl from the medium) is present in the system, which alters the course of the reaction. The solution containing Pb was mixed with apatite or PG in a medium containing Cl and sterilized. This is the step at which most of the Pb was probably removed (despite using fluorapatite which is the least effective in Pb removal. The most effective is hydroxylapatite which has been compared and is also well described in the literature introducing Pb-immobilization). Unfortunately, the authors not only failed to check the Pb concentration after sterilization and before microbial inoculation, but also failed to examine these solids by SEM before adding microorganisms. This is a manifestation of the failure to use a correct blank in the experiment, which led to incorrect conclusions in this work.

3) SEM microscopy was used in secondary electrons mode instead for backscattered electron mode. This prevented the identification of PYR on microscopic images. This is another evidence of lack of understanding of the principles.  

The results:

This study lacks of a proper control sample: Pb-contaminated solution mixed with source of P (apatite or PG) in the medium, reacted for 2, 4 and 6 days in the absence of microorganisms.

Some examples of language issues (despite the fact that the reviewer doesn’t feel qualified to judge about the English language and style)

Line 47

In the case of fluorapatite (FAp), the most common phosphate rock in nature is  effectively in Pb remediation via the formation of Pryo

Line 52

However, most P in PG is existed as the insoluble phosphate [16,17].

Line 132

2 g principles and 40 mL extractant were mixed and shaken at 180 rpm, 25 °C for 18 h.

Figure 4

The Pb concentration in adding FAp (A) and PG (B) treatments during the incubation

Reviewer 2 Report

Strength:

The manuscript deals with the Remediation of lead contaminations by Aspergillus Niger and phosphate rocks under different nitrogen. This helps decision makers for bioremediation measures and scientist to carry forward the process of developing decision support system.

Weakness

Lack of statistical analysis and weak discussion section. Line 113: Experimental design descitption and statistical analysis

Before conclusion: Where is statistical analysis?

Discussion: Lack of recent supporting and contrasting literature

Conclusion: Lack challenges for future and path ahead.

References: please cross check all references